# Spike-driven Transformer

**Man Yao**[1,2][*]**, Jiakui Hu**[3,1*]**, Zhaokun Zhou**[3,2*]**, Li Yuan**[3,2]**,**
**Yonghong Tian**[3,2]**, Bo Xu**[1]**, Guoqi Li**[1][†]

[1]Institute of Automation, Chinese Academy of Sciences, Beijing, China
[2]Peng Cheng Laboratory, Shenzhen, Guangzhou, China
[3]Peking University, Beijing, China

## Abstract

Spiking Neural Networks (SNNs) provide an energy-efficient deep learning option due to their unique spike-based event-driven (i.e., spike-driven) paradigm. In this paper, we incorporate the spike-driven paradigm into Transformer by the proposed Spike-driven Transformer with *four* unique properties: i) Event-driven, no calculation is triggered when the input of Transformer is zero; ii) Binary spike communication, all matrix multiplications associated with the spike matrix can be transformed into sparse additions; iii) Self-attention with linear complexity at both token and channel dimensions; iv) The operations between spike-form Query, Key, and Value are mask and addition. Together, there are only sparse addition operations in the Spike-driven Transformer. To this end, we design a novel Spike-Driven Self-Attention (SDSA), which exploits only mask and addition operations without any multiplication, and thus having up to $87.2\times$ lower computation energy than vanilla self-attention. Especially in SDSA, the matrix multiplication between Query, Key, and Value is designed as the mask operation. In addition, we rearrange all residual connections in the vanilla Transformer before the activation functions to ensure that all neurons transmit binary spike signals. It is shown that the Spike-driven Transformer can achieve 77.1% top-1 accuracy on ImageNet-1K, which is the state-of-the-art result in the SNN field. All source code and models are available at `https://github.com/BICLab/Spike-Driven-Transformer`.

## 1 Introduction

One of the most crucial computational characteristics of bio-inspired Spiking Neural Networks (SNNs) [1] is spike-based event-driven (spike-driven): i) When a computation is event-driven, it is triggered sparsely as events (spike with address information) occur; ii) If only binary spikes (0 or 1) are employed for communication between spiking neurons, the network's operations are synaptic ACcumulate (AC). When implementing SNNs on neuromorphic chips, such as TrueNorth [2], Loihi [3], and Tianjic [4], only a small fraction of spiking neurons at any moment being active and the rest being idle. Thus, spike-driven neuromorphic computing that only performs sparse addition operations is regarded as a promising low-power alternative to traditional AI [5, 6, 7].

Although SNNs have apparent advantages in bio-plausibility and energy efficiency, their applications are limited by poor task accuracy. Transformers have shown high performance in various tasks for their self-attention [8, 9, 10]. Incorporating the effectiveness of Transformer with the high energy efficiency of SNNs is a natural and exciting idea. There has been some research in this direction, but all so far have relied on "hybrid computing". Namely, Multiply-and-ACcumulate (MAC) operations dominated

---

[*]Equal contribution
[†]Corresponding author, guoqi.li@ia.ac.cn

37th Conference on Neural Information Processing Systems (NeurIPS 2023).

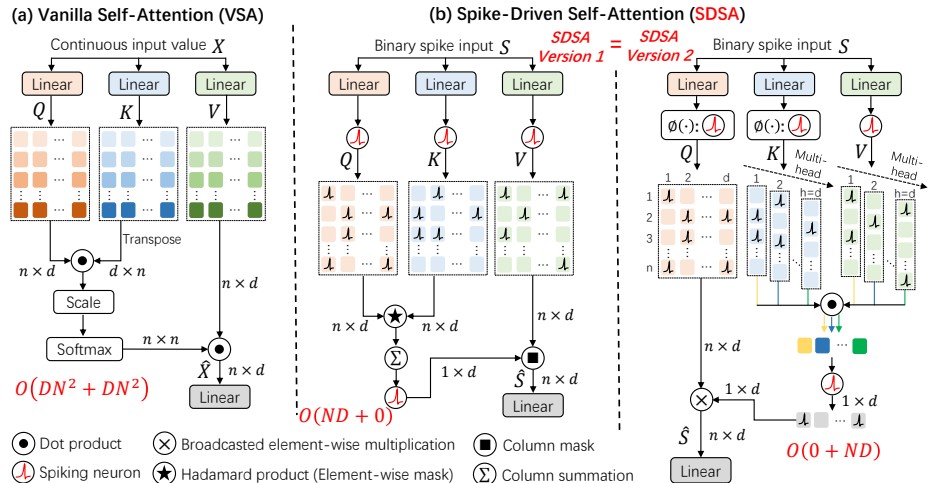

Figure 1: Comparison Vanilla Self-Attention (VSA) and our Spike-Driven Self-Attention (SDSA). **(a)** is a typical Vanilla Self-Attention (VSA) [8]. **(b)** are two equivalent versions of SDSA. The input of SDSA are binary spikes. In SDSA, there are only mask and sparse additions. Version 1: Spike $Q$ and $K$ first perform element-wise mask, i.e., Hadamard product; then column summation and spike neuron layer are adopted to obtain the binary attention vector; finally, the binary attention vector is applied to the spike $V$ to mask some channels (features). Version 2: An equivalent version of Version 1 (see Section 3.3) reveals that SDSA is a unique type of linear attention (spiking neuron layer is the kernel function) whose time complexity is linear with both token and channel dimensions. Typically, performing self-attention in VSA and SDSA requires $2N^2D$ multiply-and-accumulate and $0.02ND$ accumulate operations respectively, where $N$ is the number of tokens, $D$ is the channel dimensions, $0.02$ is the non-zero ratio of the matrix after the mask of $Q$ and $K$. Thus, the self-attention operator between the spike $Q$, $K$, and $V$ has almost no energy consumption.

by vanilla Transformer components and AC operations caused by spiking neurons both exist in the existing spiking Transformers. One popular approach is to replace some of the neurons in Transformer with spiking neurons to do a various of tasks [11, 12, 13, 14, 15, 16, 17, 18, 19, 20, 21, 22], and keeping the MAC-required operations like dot-product, softmax, scale, etc.

Though hybrid computing helps reduce the accuracy loss brought on by adding spiking neurons to the Transformer, it can be challenging to benefit from SNN's low energy cost, especially given that current spiking Transformers are hardly usable on neuromorphic chips. To address this issue, we propose a novel Spike-driven Transformer that achieves the spike-driven nature of SNNs throughout the network while having great task performance. Two core modules of Transformer, Vanilla Self-Attention (VSA) and Multi-Layer Perceptron (MLP), are re-designed to have a spike-driven paradigm.

The three input matrices for VSA are Query ($Q$), Key ($K$), and Value ($V$), (Fig. 1(a)). $Q$ and $K$ first perform similarity calculations to obtain the attention map, which includes three steps of matrix multiplication, scale and softmax. The attention map is then used to weight the $V$ (another matrix multiplication). The typical spiking self-attentions in the current spiking Transformers [20, 19] would convert $Q$, $K$, $V$ into spike-form before performing two matrix multiplications similar to those in VSA. The distinction is that spike matrix multiplications can be converted into addition, and softmax is not necessary [20]. But these methods not only yield large integers in the output (thus requiring an additional scale multiplication for normalization to avoid gradient vanishing), but also fails to exploit the full energy-efficiency potential of the spike-driven paradigm combined with self-attention.

We propose Spike-Driven Self-Attention (SDSA) to address these issues, including two aspects (see SDSA Version 1 in Fig. 1(b)): i) Hadamard product replaces matrix multiplication; ii) matrix column-wise summation and spiking neuron layer take the role of softmax and scale. The former can be considered as not consuming energy because the Hadamard product between spikes is equivalent to the element-wise mask. The latter also consumes almost no energy since the matrix to be summed column-by-column is very sparse (typically, the ratio of non-zero elements is less than 0.02). We also observe that SDSA is a special kind of linear attention [23, 24], i.e., Version 2 of Fig. 1(b). In this view, the spiking neuron layer that converts $Q$, $K$, and $V$ into spike form is a kernel function.

Additionally, existing spiking Transformers [20, 12] usually follow the SEW-SNN residual design [25], whose shortcut is spike addition and thus outputs multi-bit (integer) spikes. This shortcut can satisfy event-driven, but introduces integer multiplication. We modified the residual connections throughout the Transformer architecture as shortcuts between membrane potentials [26, 27] to address this issue (Section 3.2). The proposed Spike-driven Transformer improves task accuracy on both static and neuromorphic event-based datasets. The main contributions of this paper are as follows:

- We propose a novel Spike-driven Transformer that only exploits sparse addition. This is the first time that the spike-driven paradigm has been incorporated into Transformer, and the proposed model is hardware-friendly to neuromorphic chips.

- We design a Spike-Driven Self-Attention (SDSA). The self-attention operator between spike Query, Key, Value is replaced by mask and sparse addition with essentially no energy consumption. SDSA is computationally linear in both tokens and channels. Overall, the energy cost of SDSA (including Query, Key, Value generation parts) is $87.2\times$ lower than its vanilla self-attention counterpart.

- We rearrange the residual connections so that all spiking neurons in Spike-driven Transformer communicate via binary spikes.

- Extensive experiments show that the proposed architecture can outperform or comparable to State-Of-The-Art (SOTA) SNNs on both static and neuromorphic datasets. We achieved 77.1% accuracy on ImageNet-1K, which is the SOTA result in the SNN field.

## 2 Related Works

**Bio-inspired Spiking Neural Networks** can profit from advanced deep learning and neuroscience knowledge concurrently [28, 5, 7]. Many biological mechanisms are leveraged to inspire SNN's neuron modeling [1, 29], learning rules [30, 31], etc. Existing studies have shown that SNNs are more suited for incorporating with brain mechanisms, e.g., long short-term memory [32, 33], attention [34, 27, 35], etc. Moreover, while keeping its own spike-driven benefits, SNNS have greatly improved its task accuracy by integrating deep learning technologies like network architecture [26, 25, 36], gradient backpropagation [37, 38], normalization [39, 40], etc. Our goal is to combine SNN and Transformer architectures. One way is to discretize Transformer into spike form through neuron equivalence [41, 42], i.e., ANN2SNN, but this requires a long simulation timestep and boosts the energy consumption. We employ the direct training method, using the first SNN layer as the spike encoding layer and applying surrogate gradient training [43].

**Neuromorphic Chips.** As opposed to the compute and memory separated processors used in ANNs, neuromorphic chips use non-von Neumann architectures, which are inspired by the structure and function of the brain [5, 7, 28]. Because of the choice that uses spiking neurons and synapses as basic units, neuromorphic chips [44, 45, 46, 2, 47, 3, 4] have unique features, such as highly parallel operation, collocated processing and memory, inherent scalability, and spike-driven computing, etc. Typical neuromorphic chips consume tens to hundreds of mWs [48]. Conv and MLP in neuromorphic chips are equivalent to a cheap addressing algorithm [49] since the sparse spike-driven computing, i.e., to find out which synapses and neurons need to be involved in the addition operation. Our Transformer design strictly follows the spike-driven paradigm, thus it is friendly to deploy on neuromorphic chips.

**Efficient Transformers.** Transformer and its variants have been widely used in numerous tasks, such as natural language processing [8, 50, 51] and computer vision [52, 53, 54]. However, deploying these models on mobile devices with limited resources remains challenging because of their inherent complexity [24, 55]. Typical optimization methods include convolution and self-attention mixing [55, 56], Transformer's own mechanism (token mechanism [57, 58], self-attention [59, 60], multi-head [61, 62] and so on) optimization, etc. An important direction for efficient Transformers is linear attention on tokens since the computation scale of self-attention is quadratic with token number. Removing the softmax in self-attention and re-arranging the computation order of Query, Key, and Value is the main way to achieve linear attention [60, 63, 23, 64, 65, 62, 66]. For the spiking Transformer, softmax cannot exist, thus spiking Transformer can be a kind of linear attention.

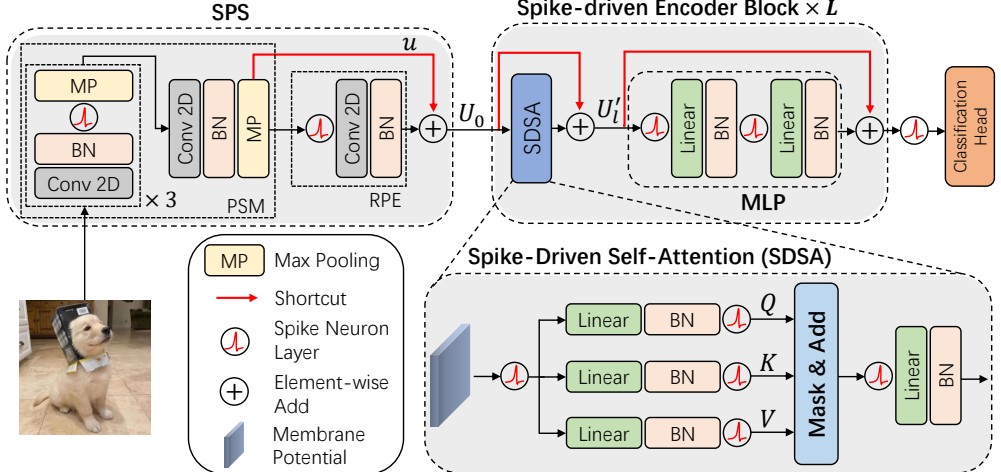

Figure 2: The overview of Spike-driven Transformer. We follow the network structure in [20], but make two new key designs. First, we propose a Spike-driven Self-Attention (SDSA) module, which consists of only mask and sparse addition operations (Fig. 1(b)). Second, we redesign the shortcuts in the whole network, involving position embedding, self-attention, and MLP parts. As indicated by the red line, the shortcut is constructed before the spike neuron layer. That is, we establish residual connections between membrane potentials to make sure that the values in the spike matrix are all binary, which allows the multiplication of the spike and weight matrices to be converted into addition operations. By contrast, previous works [20, 19] build shortcut between spike tensors in different layers, resulting in the output of spike neurons as multi-bit (integer) spikes.

## 3 Spike-driven Transformer

We propose a Spike-driven Transformer, which incorporates Transformer into the spike-driven paradigm with only sparse addition. We first briefly introduce the spike neuron layer, then introduce the overview and components of the Spike-driven Transformer one by one.

The spiking neuron model is simplified from the biological neuron model [67, 29]. Leaky Integrate-and-Fire (LIF) spiking neuron [1], which have biological neuronal dynamics and are easy to simulate on a computer, is uniformly adopted in this work. The dynamics of the LIF layer [37] is governed by

$$U[t] = H[t-1] + X[t], \tag{1}$$

$$S[t] = \text{Hea}\left(U[t] - u_{th}\right), \tag{2}$$

$$H[t] = V_{reset}S[t] + (\beta U[t])\left(\mathbf{1} - S[t]\right), \tag{3}$$

where $t$ denote the timestep, $U[t]$ means the membrane potential which is produced by coupling the spatial input information $X[t]$ and the temporal input $H[t-1]$, where $X[t]$ can be obtained through operators such as Conv, MLP, and self-attention. When membrane potential exceeds the threshold $u_{th}$, the neuron will fire a spike, otherwise it will not. Thus, the spatial output tensor $S[t]$ contains only 1 or 0. $\text{Hea}(\cdot)$ is a Heaviside step function that satisfies $\text{Hea}(x) = 1$ when $x \geq 0$, otherwise $\text{Hea}(x) = 0$. $H[t]$ indicates the temporal output, where $V_{reset}$ denotes the reset potential which is set after activating the output spiking. $\beta < 1$ is the decay factor, if the spiking neuron does not fire, the membrane potential $U[t]$ will decay to $H[t]$.

### 3.1 Overall Architecture

Fig. 2 shows the overview of Spike-driven Transformer that includes four parts: Spiking Patch Splitting (SPS), SDSA, MLP, and a linear classification head. For the SPS part, we follow the design in [20]. Given a 2D image sequence $I \in \mathbb{R}^{T \times C \times H \times W}$, the Patch Splitting Module (PSM), i.e., the first four Conv layers, linearly projects and splits it into a sequence of $N$ flattened spike patches $s$ with $D$ dimensional channel, where $T$ (images are repeated $T$ times in the static dataset as input), $C$, $H$, and $W$ denote timestep, channel, height and width of the 2D image sequence. Another Conv layer

is then used to generate Relative Position Embedding (RPE). Together, the SPS part is written as:

$$u = \text{PSM}\,(I), \qquad\qquad I \in \mathbb{R}^{T \times C \times H \times W}, u \in \mathbb{R}^{T \times N \times D} \tag{4}$$

$$s = \mathcal{SN}(u), \qquad\qquad s \in \mathbb{R}^{T \times N \times D} \tag{5}$$

$$\text{RPE} = \text{BN}(\text{Conv2d}(s)), \qquad\qquad \text{RPE} \in \mathbb{R}^{T \times N \times D} \tag{6}$$

$$U_0 = u + \text{RPE}, \qquad\qquad U_0 \in \mathbb{R}^{T \times N \times D} \tag{7}$$

where $u$ and $U_0$ are the output membrane potential tensor of PSM and SPS respectively, $\mathcal{SN}(\cdot)$ denote the spike neuron layer. Note, in Eq 6, before executing $Conv2d(\cdot)$, $s \in \mathbb{R}^{T \times N \times D}$ will be transposed into $s \in \mathbb{R}^{T \times C \times H \times W}$. We pass the $U_0$ to the $L$-block Spike-driven Transformer encoder, which consists of a SDSA and a MLP block. Residual connections are applied to membrane potentials in both SDSA and MLP block. SDSA provides an efficient approach to model the local-global information of images utilizing spike $Q$, $K$, and $V$ without scale and softmax (see Sec. 3.3). A Global Average-Pooling (GAP) is utilized on the processed feature from spike-driven encoder and outputs the $D$-dimension channel which will be sent to the fully-connected-layer Classification Head (CH) to output the prediction $Y$. The three parts SDSA, MLP and CH can be written as follows:

$$S_0 = \mathcal{SN}(U_0), \qquad\qquad S_0 \in \mathbb{R}^{T \times N \times D} \tag{8}$$

$$U_l^{'} = \text{SDSA}(S_{l-1}) + U_{l-1}, \qquad\qquad U_l^{'} \in \mathbb{R}^{T \times N \times D}, l = 1...L \tag{9}$$

$$S_l^{'} = \mathcal{SN}(U_l^{'}), \qquad\qquad S_l^{'} \in \mathbb{R}^{T \times N \times D}, l = 1...L \tag{10}$$

$$S_l = \mathcal{SN}(\text{MLP}(S_l^{'}) + U_l^{'}), \qquad\qquad S_l \in \mathbb{R}^{T \times N \times D}, l = 1...L \tag{11}$$

$$Y = \text{CH}(\text{GAP}(S_L)), \tag{12}$$

where $U_l^{'}$ and $S_l^{'}$ represent membrane potential and spike tensor output by SDSA at $l$-th layer.

## 3.2 Membrane Shortcut in Spike-driven Transformer

Residual connection [68, 69] is a crucial basic operation in Transformer architecture. There are three shortcut techniques in existing Conv-based SNNs [27]. Vanilla Res-SNN [39], similar to vanilla Res-CNN [68], performs a shortcut between membrane potential and spike. Spike-Element-Wise (SEW) Res-SNN [25] employs a shortcut to connect the output spikes in different layers. Membrane Shortcut (MS) Res-SNN [26], creating a shortcut between membrane potential of spiking neurons in various layers. There is no uniformly standard shortcut in the current SNN community, and SEW shortcut is adopted by existing spiking Transformers [20, 12]. As shown in Eq. 7, Eq. 9 and Eq. 11, we leverage the membrane shortcut in the proposed Spike-driven Transformer for four reasons:

- *Spike-driven* refers to the ability to transform matrix multiplication between weight and spike tensors into sparse additions. Only binary spikes can support the spike-driven function. However, the values in the spike tensors are multi-bit (integer) spikes, as the SEW shortcut builds the addition between binary spikes. By contrast, as shown in Eq. 8, Eq. 10, Eq. 11, $\mathcal{SN}$ is followed by the MS shortcut, which ensures that there are always only binary spike signals in the spike tensor.

- *High performance*. The task accuracy of MS-Res-SNN is higher than that of SEW-Res-SNN [26, 27, 25], also in Transformer-based SNN (see Table 5 in this work).

- *Bio-plausibility*. MS shortcut can be understood as an approach to optimize the membrane potential distribution. This is consistent with other neuroscience-inspired methods to optimize the internal dynamics of SNNs, such as complex spiking neuron design [70], attention mechanism [27], long short-term memory [33], recurrent connection [71], information maximization [72], etc.

- *Dynamical isometry*. MS-Res-SNN has been proven [26] to satisfy dynamical isometry theory [73], which is a theoretical explanation of well-behaved deep neural networks [74, 75].

## 3.3 Spike-driven Self-Attention

**Vanilla Self-Attention (VSA).** Given a float-point input feature sequence $X \in \mathbb{R}^{N \times D}$, float-point Query ($Q$), Key ($K$), and Value ($V$) in $\mathbb{R}^{N \times D}$ are calculated by three learnable linear matrices, respectively. Standard scaled dot-product self-attention (Fig. 2(a)) is computed as [52]:

$$\text{VSA}(Q, K, V) = \text{softmax}\left(\frac{QK^{\text{T}}}{\sqrt{d}}\right) V, \tag{13}$$

where $d = D/H$ is the feature dimension of one head and $H$ is the head number, $\sqrt{d}$ is the scale factor. The time complexity of VSA is $O(N^2 D + N^2 D)$.

**Spike-Driven Self-Attention (SDSA) Version 1.** As shown in Fig. 2(b) left part, given a spike input feature sequence $S \in \mathbb{R}^{T \times N \times D}$, float-point $Q$, $K$, and $V$ in $\mathbb{R}^{T \times N \times D}$ are calculated by three learnable linear matrices, respectively. Note, the linear operation here is only addition, because the input $S$ is a spike tensor. A spike neuron layer $\mathcal{SN}(\cdot)$ follows, converting $Q$, $K$, $V$ into spike tensor $Q_S$, $K_S$, and $V_S$. SDSA Version 1 (SDSA-V1) is presented as:

$$\text{SDSA}(Q, K, V) = g(Q_S, K_S) \otimes V_S = \mathcal{SN}\left(\text{SUM}_\text{c}\left(Q_S \otimes K_S\right)\right) \otimes V_S, \tag{14}$$

where $\otimes$ is the Hadamard product, $g(\cdot)$ is used to compute the attention map, $\text{SUM}_\text{c}(\cdot)$ represents the sum of each column. The outputs of both $g(\cdot)$ and $\text{SUM}_\text{c}(\cdot)$ are $D$-dimensional row vectors. The Hadamard product between spike tensors is equivalent to the mask operation.

**Discussion on SDSA.** Since the Hadamard product among $Q_S$, $K_S$, and $V_S$ in $\mathbb{R}^{N \times D}$ (we here assume $T = 1$ for mathematical understanding) can be exchanged, Eq. 14 can also be written as:

$$\text{SDSA}(Q, K, V) = Q_S \otimes g(K_S, V_S) = Q_S \otimes \mathcal{SN}\left(\text{SUM}_\text{c}\left(K_S \otimes V_S\right)\right). \tag{15}$$

Note, Eq.14 and Eq.15 are equivalent functionally. In this view, Eq.15 is a linear attention [23, 63] whose computational complexity is linear in token number $N$ because $K_S$ and $V_S$ can participate in calculate first. This is thanks to the softmax operation in the VSA is dropped here. The function of softmax needs to be replaced by the kernel function. Specific to our SDSA, $\mathcal{SN}(\cdot)$ is the kernel function. Further, we can assume a special case [62], $H = D$, i.e., the number of channels per head is one. After the self-attention operation is performed on the $H$ heads respectively, the outputs are concatenated together. Specifically,

$$\text{SDSA}(Q^i, K^i, V^i) = \mathcal{SN}(Q^i)g(K^i, V^i) = \mathcal{SN}(Q^i)\mathcal{SN}\left(\mathcal{SN}(Q^i)^\text{T} \odot \mathcal{SN}(V^i)\right), \tag{16}$$

where $Q^i, K^i, V^i$ in $\mathbb{R}^{N \times 1}$ are the $i$-th vectors in $Q, K, V$ respectively, $\odot$ is the dot product operation. The output of $g(K^i, V^i)$ is a scalar, 0 or 1. Since the operation between $\mathcal{SN}(Q^i)$ and $g(K^i, V^i)$ is a mask, the whole SDSA only needs to be calculated $H = D$ times for $g(K^i, V^i)$. The computational complexity of SDSA is $O(0 + ND)$, which is linear with both $N$ and $D$ (see Fig. 1(b) right part). Vectors $K^i$ and $V^i$ are very sparse, typically less than 0.01 (Table 2). Together, the whole SDSA only needs about $0.02ND$ times of addition, and its energy consumption is negligible.

Interestingly, Eq. 15 actually converts the soft vanilla self-attention to hard self-attention, where the attention scores in soft and hard attention are continuous- and binary-valued, respectively [76]. Thus, the practice of the spike-driven paradigm in this work leverages binary self-attention scores to directly mask unimportant channels in the sparse spike Value tensor. Although this introduces a slight loss of accuracy (Table 5), $\text{SDSA}(\cdot)$ consumes almost no energy.

## 4 Theoretical Energy Consumption Analysis

Three key computational modules in deep learning are Conv, MLP, and self-attention. In this Section, We discuss how the spike-driven paradigm achieves high energy efficiency on these operators.

**Spike-driven in Conv and MLP.** Spike-driven combines two properties, event-driven and binary spike-based communication. The former means that no computation is triggered when the input is zero. The binary restriction in the latter indicates that there are only additions. In summary, in spike-driven Conv and MLP, matrix multiplication is transformed into sparse addition, which is implemented as addressable addition in neuromorphic chips [49].

**Spike-driven in Self-attention.** $Q_S$, $K_S$, $V_S$ in spiking self-attention involve two matrix multiplications. One approach is to perform multiplication directly between $Q_S$, $K_S$, $V_S$, which is then converted to sparse addition, like spike-driven Conv and MLP. The previous work [20] did just that. We provide a new scheme that performs element-wise multiplication between $Q_S$, $K_S$, $V_S$. Since all elements in spike tensors are either 0 or 1, element multiplication is equivalent to a mask operation with no energy consumption. Mask operations can be implemented in neuromorphic chips through addressing algorithms [49] or AND logic operations [4].

**Energy Consumption Comparison.** The times of floating-point operations (FLOPs) is often used to estimate the computational burden in ANNs, where almost all FLOPs are MAC. Under the same

Table 1: Energy evaluation. $FL_{Conv}$ and $FL_{MLP}$ represent the FLOPs of the Conv and MLP models in the ANNs, respectively. $R_C$, $R_M$, $\overline{R}$, $\widehat{R}$ denote the spike firing rates (the proportion of non-zero elements in the spike matrix) in various spike matrices. We give the strict definitions and calculation methods of these indicators in the Supplementary due to space constraints.

|  |  | Vanilla Transformer [52] | Spike-driven Transformer (**This work**) |
|---|---|---|---|
| SPS | First Conv | $E_{MAC} \cdot FL_{Conv}$ | $E_{MAC} \cdot T \cdot R_C \cdot FL_{Conv}$ |
|  | Other Conv | $E_{MAC} \cdot FL_{Conv}$ | $E_{AC} \cdot T \cdot R_C \cdot FL_{Conv}$ |
| Self-attention | $Q, K, V$ | $E_{MAC} \cdot 3ND^2$ | $E_{AC} \cdot T \cdot \overline{R} \cdot 3ND^2$ |
|  | $f(Q, K, V)$ | $E_{MAC} \cdot 2N^2 D$ | $E_{AC} \cdot T \cdot \widehat{R} \cdot ND$ |
|  | Scale | $E_M \cdot N^2$ | - |
|  | Softmax | $E_{MAC} \cdot 2N^2$ | - |
|  | Linear | $E_{MAC} \cdot FL_{MLP0}$ | $E_{AC} \cdot T \cdot R_{M0} \cdot FL_{MLP0}$ |
| MLP | Layer 1 | $E_{MAC} \cdot FL_{MLP1}$ | $E_{AC} \cdot T \cdot R_{M1} \cdot FL_{MLP1}$ |
|  | Layer 2 | $E_{MAC} \cdot FL_{MLP2}$ | $E_{AC} \cdot T \cdot R_{M2} \cdot FL_{MLP2}$ |

architecture, the energy cost of SNN can be estimated by combining the spike firing rate $R$ and simulation timestep $T$ if the FLOPs of ANN is known. Table 1 shows the energy consumption of Conv, self-attention, and MLP modules of the same scale in vanilla and our Spike-driven Transformer.

## 5 Experiments

We evaluate our method on both static datasets ImageNet [77], CIFAR-10/100 [78], and neuromorphic datasets CIFAR10-DVS [79], DVS128 Gesture [80].

**Experimental Setup on ImageNet.** For the convenience of comparison, we generally continued the experimental setup in [20]. The input size is set to $224 \times 224$. The batch size is set to 128 or 256 during 310 training epochs with a cosine-decay learning rate whose initial value is 0.0005. The optimizer is Lamb. The image is divided into $N = 196$ patches using the SPS module. Standard data augmentation techniques, like random augmentation, mixup, are also employed in training. Details of the training and experimental setup on ImageNet are given in the supplementary material.

**Accuracy analysis on ImageNet.** Our experimental results on ImageNet are given in Table 3. We first compare our model performance with the baseline spiking Transformer (i.e., SpikFormer [20]). The five network architectures consistent with those in SpikFormer are adopted by this work. We can see that under the same parameters, our accuracies are significantly better than the corresponding baseline models. For instance, the Spike-driven Transformer-8-384 is 2.0% higher than SpikFormer-8-384. It is worth noting that the Spike-driven Transformer-8-768 obtains 76.32% (input 224×224) with 66.34M, which is 1.5% higher than the corresponding SpikFormer. We further expand the inference resolution to 288×288, obtaining 77.1%, which is the SOTA result of the SNN field on ImageNet.

We then compare our results with existing Res-SNNs. Whether it is vanilla Res-SNN [39], MS-Res-SNN [26] or SEW-Res-SNN [25], the accuracy of Spike-driven Transformer-8-768 (77.1%) is the highest. Att-MS-Res-SNN [27] also achieves 77.1% accuracy by plugging an additional attention auxiliary module [81, 82] in MS-Res-SNN, but it destroys the spike-driven nature and requires more parameters (78.37M vs. 66.34M) and training time (1000epoch vs. 310epoch). Furthermore, the proposed Spike-

Table 2: Spike Firing Rate (SFR) of Spike-driven Self-attention in 8-512. Average SFR is the mean of SFR over $T = 4$, and 8 SDSA blocks.

| SDSA | Average SFR |
|---|---|
| $Q_S$ | 0.0091 |
| $K_S$ | 0.0090 |
| $g(Q_S, K_S)$ | 0.0713 |
| $V_S$ | 0.1649 |
| Output of SDSA$(\cdot)$, $\hat{V}_S$ | 0.0209 |

driven Transformer outperforms by more than 72% at various network scales, while Res-SNNs have lower performance with a similar amount of parameters. For example, **Spike-driven Transformer-6-512 (This work)** vs. SEW-Res-SNN-34 vs. MS-Res-SNN-34: Param, **23.27M** vs. 21.79M vs. 21.80M; Acc, **74.11%** vs. 67.04% vs. 69.15%.

Table 3: Evaluation on ImageNet. Power is the average theoretical energy consumption when predicting an image from the test set. The power data in this work is evaluated according to Table 1, and data for other works were obtained from related papers. Spiking Transformer-$L$-$D$ represents a model with $L$ encoder blocks and $D$ channels. *The input crops are enlarged to 288×288 in inference. The default inference input resolution for other models is 224×224.

| Methods | Architecture | Spike -driven | Param (M) | Power (mJ) | Time Step | Acc |
|---|---|---|---|---|---|---|
| Hybrid training [83] | ResNet-34 | ✓ | 21.79 | - | 250 | 61.48 |
| TET [84] | SEW-ResNet-34 | ✗ | 21.79 | - | 4 | 68.00 |
| Spiking ResNet [85] | ResNet-50 | ✓ | 25.56 | 70.93 | 350 | 72.75 |
| tdBN [39] | Spiking-ResNet-34 | ✓ | 21.79 | 6.39 | 6 | 63.72 |
| | SEW-ResNet-34 | ✗ | 21.79 | 4.04 | 4 | 67.04 |
| SEW ResNet [25] | SEW-ResNet-50 | ✗ | 25.56 | 4.89 | 4 | 67.78 |
| | SEW-ResNet-101 | ✗ | 44.55 | 8.91 | 4 | 68.76 |
| | SEW-ResNet-152 | ✗ | 60.19 | 12.89 | 4 | 69.26 |
| | MS-ResNet-18 | ✓ | 11.69 | 4.29 | 4 | 63.10 |
| MS ResNet [26] | MS-ResNet-34 | ✓ | 21.80 | 5.11 | 4 | 69.42 |
| | MS-ResNet-104* | ✓ | 77.28 | 10.19 | 4 | 76.02 |
| | Att-MS-ResNet-18 | ✗ | 11.87 | 0.48 | 1 | 63.97 |
| Att MS ResNet [27] | Att-MS-ResNet-34 | ✗ | 22.12 | 0.57 | 1 | 69.15 |
| | Att-MS-ResNet-104* | ✗ | 78.37 | 7.30 | 4 | **77.08** |
| ResNet | Res-CNN-104 | ✗ | 77.28 | 54.21 | 1 | 76.87 |
| Transformer | Transformer-8-512 | ✗ | 29.68 | 41.77 | 1 | **80.80** |
| | Spiking Transformer-8-384 | ✗ | 16.81 | 7.73 | 4 | 70.24 |
| | Spiking Transformer-6-512 | ✗ | 23.37 | 9.41 | 4 | 72.46 |
| Spikformer [20] | Spiking Transformer-8-512 | ✗ | 29.68 | 11.57 | 4 | 73.38 |
| | Spiking Transformer-10-512 | ✗ | 36.01 | 13.89 | 4 | 73.68 |
| | Spiking Transformer-8-768 | ✗ | 66.34 | 21.47 | 4 | 74.81 |
| | Spiking Transformer-8-384 | ✓ | 16.81 | 3.90 | 4 | 72.28 |
| | Spiking Transformer-6-512 | ✓ | 23.37 | 3.56 | 4 | 74.11 |
| **Spike-driven** | Spiking Transformer-8-512 | ✓ | 29.68 | **1.13** | 1 | 71.68 |
| **Transformer (Ours)** | Spiking Transformer-8-512 | ✓ | 29.68 | 4.50 | 4 | 74.57 |
| | Spiking Transformer-10-512 | ✓ | 36.01 | 5.53 | 4 | 74.66 |
| | Spiking Transformer-8-768* | ✓ | 66.34 | 6.09 | 4 | **77.07** |

**Power analysis on ImageNet.** Compared with prior works, the Spike-driven Transformer shines in energy cost (Table 3). We first make an intuitive comparison of energy consumption in the SNN field. **Spike-driven Transformer-8-512 (This work)** vs. SEW-Res-SNN-50 vs. MS-Res-SNN-34: Power, **4.50mJ** vs. 4.89mJ vs. 5.11mJ; Acc, **74.57%** vs. 67.78% vs. 69.42%. That is, our model achieves +6.79% and +5.15% accuracy higher than previous SEW and MS Res-SNN backbones with lower energy consumption. What is more attractive is that the energy efficiency of the Spike-driven Transformer will be further expanded as the model scale grows because its computational complexity is linear in both token and channel dimensions. For instance, in an 8-layer network, as the channel dimension increases from **384** to **512** and **768**, SpikFormer [20] has **1.98**×(7.73mJ/3.90mJ), **2.57**×(11.57mJ/4.50mJ), and **3.52**×(21.47mJ/6.09mJ) higher energy consumption than our Spike-driven Transformer. At the same time, our task performance on these three network structures has improved by **+2.0%**, **+1.2%**, and **+1.5%**, respectively.

Then we compare the energy cost between Spike-driven and ANN Transformer. Under the same structure, such as 8-512, the power required by the ANN-ViT (41.77mJ) is **9.3**× that of the spike-driven counterpart (4.50mJ). Further, the energy advantage will extend to **36.7**× if we set $T = 1$ in the Spike-driven version (1.13mJ). Although the accuracy of $T = 1$ (here we are direct training) will be lower than $T = 4$, it can be compensated

Table 4: Energy Consumption of Self-attention. $E_1$ and $E_2$ (including energy consumption to generate $Q, K, V$) represent the power of self-attention mechanism in ANN and spike-driven.

| Models | $E_1$ (pJ) | $E_2$ (pJ) | $E_1/E_2$ |
|---|---|---|---|
| 8-384 | 6.7e8 | 1.6e7 | **42.6** |
| 8-512 | 1.2e9 | 2.1e7 | **57.2** |
| 8-768 | 2.7e9 | 3.1e7 | **87.2** |

Table 5: Experimental Results on CIFAR10/100, DVS128 Gesture and CIFAR10-DVS.

| Methods | Spike-driven | CIFAR10-DVS | | DVS128 Gesture | | CIFAR-10 | | CIFAR-100 | |
|---|---|---|---|---|---|---|---|---|---|
| | | $T$ | Acc | $T$ | Acc | $T$ | Acc | $T$ | Acc |
| tdBN [39] | ✗ | 10 | 67.8 | 40 | 96.9 | 6 | 93.2 | - | - |
| PLIF [87] | ✓ | 20 | 74.8 | 20 | 97.6 | 8 | 93.5 | - | - |
| Dspike [88] | ✗ | 10 | 75.4 | - | - | 6 | 94.3 | 6 | 74.2 |
| DSR [89] | ✓ | 10 | 77.3 | - | - | 20 | 95.4 | 20 | **78.5** |
| Spikformer [20] | ✗ | 16 | **80.9** | 16 | 98.3 | 4 | 95.5 | 4 | 78.2 |
| DIET-SNN[90] | ✗ | - | - | - | - | 5 | 92.7 | 5 | 69.7 |
| ANN(ResNet19) | ✗ | - | - | - | - | 1 | 94.97 | 1 | 75.4 |
| ANN(Transformer4-384) | ✗ | - | - | - | - | 1 | 96.7 | 1 | 81.0 |
| **This Work** | ✓ | 16 | 80.0 | 16 | **99.3** | 4 | **95.6** | 4 | 78.4 |

by special training methods [86] in future work.

Sparse spike firing is the key for Spike-driven Transformer to achieve high energy efficiency. As shown in Table 2, the Spike Firing Rate (SFR) of the self-attention part is very low, where the SFR of $Q_S$ and $Q_K$ are both less than 0.01. Since the mask (Hadamard product) operation does not consume energy, the number of additions required by the $\mathrm{SUM_c}\,(Q_S \otimes K_S)$ is less than $0.02ND$ times. The operation between the vector output by $g(Q_S, K_S)$ and $V_S$ is still a column mask that does not consume energy. Consequently, in the whole self-attention part, the energy consumption of spike-driven self-attention can be lower than $87.2\times$ of ANN self-attention (see Table 4).

**Experimental results on CIFAR-10/100, CIFAR10-DVS, and DVS128 Gesture** are conducted in Table 5. These four datasets are relatively small compared to ImageNet. CIFAR-10/100 are static image classification datasets. Gesture and CIFAR10-DVS are neuromorphic action classification datasets, which need to convert the event stream into frame sequences

Table 6: Studies on Spiking Transformer-2-512.

| Model | CIFAR-10 | CIFAR-100 |
|---|---|---|
| Baseline [20] | 93.12 | 73.17 |
| + SDSA | 93.09 (-0.03) | 72.83 (-0.34) |
| + MS | 93.93 (+0.81) | 74.63 (+1.46) |
| This work | 93.82 (+0.73) | 74.41 (+1.24) |

before processing. DVS128 Gesture is a gesture recognition dataset. CIFAR10-DVS is a neuromorphic dataset converted from CIFAR-10 by shifting image samples to be captured by the DVS camera. We basically keep the experimental setup in [20], including the network structure, training settings, etc., and details are given in the supplementary material. As shown in Table 5, we achieve SOTA results on Gesture (99.3%) and CIFAR-10 (95.6%), and comparable results to SOTA on other datasets.

**Ablation study.** To implement the spike-driven paradigm in Transformer, we design a new SDSA module and reposition the residual connections in the entire network based on the SpikFormer [20]. We organize ablation studies on CIFAR10/100 to analyze their impact. Results are given in Table 5. We adopt spiking Transformer-2-512 as the baseline structure. It can be observed that SDSA incurs a slight performance loss. As discussed in Section 3.3, SDSA actually masks some unimportant channels directly. In Fig. 3,

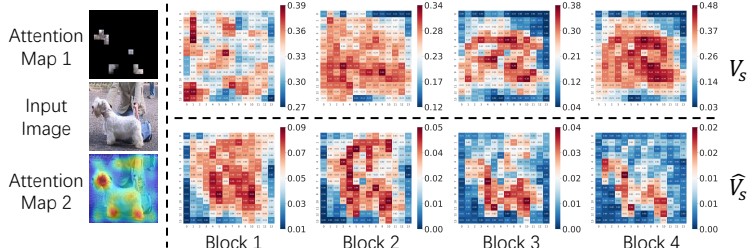

Figure 3: Attention Map Based on Spike Firing Rate (SFR). $V_S$ is the Value tensor. $\hat{V}_S$ is the output of $\mathrm{SDSA}(\cdot)$. The spike-driven self-attention mechanism masks unimportant channels in $V_S$ to obtain $\hat{V}_S$. Each pixel on $V_S$ and $\hat{V}_S$ represents the SFR at a patch. The spatial resolution of each attention map is $14 \times 14$ (196 patches). The redder the higher the SFR, the bluer the smaller the SFR.

we plot the attention maps (the detailed drawing method is given in the supplementary material), and we can observe: i) SDSA can optimize intermediate features, such as masking background information; ii) SDSA greatly reduces the spike firing rate of $\hat{V}_S$, thereby reducing energy cost. On the other hand, the membrane shortcut leads to significant accuracy improvements, consistent

with the experience of Conv-based MS-SNN [26, 25]. Comprehensively, the proposed Spike-driven Transformer simultaneously achieves better accuracy and lower energy consumption (Table 3).

## 6    Conclusion

We propose a Spike-driven Transformer that combines the low power of SNN and the excellent accuracy of the Transformer. There is only sparse addition in the proposed Spike-driven Transformer. To this end, we design a novel Spike-Driven Self-Attention (SDSA) module and rearrange the location of residual connections throughout the network. The complex and energy-intensive matrix multiplication, softmax, and scale in the vanilla self-attention are dropped. Instead, we employ mask, addition, and spike neuron layer to realize the function of the self-attention mechanism. Moreover, SDSA has linear complexity with both token and channel dimensions. Extensive experiments are conducted on static image and neuromorphic datasets, verifying the effectiveness and efficiency of the proposed method. We hope our investigations pave the way for further research on Transformer-based SNNs and inspire the design of next-generation neuromorphic chips.

## Acknowledgement

This work was supported by Beijing Natural Science Foundation for Distinguished Young Scholars (JQ21015), National Science Foundation for Distinguished Young Scholars (62325603), and National Natural Science Foundation of China (62236009, U22A20103).

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
