# S1 Energy Consumption Analysis Details

We show the theoretical energy consumption estimation method of the proposed Spike-driven Transformer in Table 1 of the main text. Compared to the vanilla Transformer counterpart, the spiking version requires information on timesteps $T$ and spike firing rates ($R$). Therefore, we only need to evaluate the FLOPs of the vanilla Transformer, and $T$ and $R$ are known, we can get the theoretical energy consumption of spike-driven Transformer.

The FLOPs of the $n$-th Conv layer in ANNs [91] are:

$$FL_{Conv} = (k_n)^2 \cdot h_n \cdot w_n \cdot c_{n-1} \cdot c_n, \tag{S1}$$

where $k_n$ is the kernel size, $(h_n, w_n)$ is the output feature map size, $c_{n-1}$ and $c_n$ are the input and output channel numbers, respectively. The FLOPs of the $m$-th MLP layer in ANNs are:

$$FL_{MLP} = i_m \cdot o_m, \tag{S2}$$

where $i_m$ and $o_m$ are the input and output dimensions of the MLP layer, respectively.

The spike firing rate is defined as the proportion of non-zero elements in the spike tensor. In Table S1, we present the spike firing rates for all spiking tensors in spike-driven Transformer-8-512. In addition, $\overline{R}$ in Table 1 indicates the average of the spike firing rates of $Q_S$, $K_S$, and $V_S$. $\widehat{R}$ is the sum of the spike firing rates of $Q_S$ and $K_S$.

Refer to previous works[92, 71, 93, 27], we assume the data for various operations are 32-bit floating point implementation in 45nm technology [94], in which $E_{MAC} = 4.6pJ$ and $E_{AC} = 0.9pJ$. Overall, for the same operator (Conv, MLP, Self-attention), as long as $E_{AC} \times T \times R < E_{MAC}$, SNNs are theoretically more energy efficient than counterpart ANNs. $E_{AC} \times T$ is usually a constant, thus sparser spikes (smaller $R$) result in lower energy cost.

# S2 Experiment Details

**Datasets.** We employ two types of datasets: static image classification and neuromorphic classification. The former includes ImageNet-1K [77], CIFAR-10/100 [78]. The latter contains CIFAR10-DVS [79] and DVS128 Gesture [80].

ImageNet-1K is the most typical static image dataset, which is widely used in the field of image classification. It offers a large-scale natural image dataset of 1.28 million training images and 50k test images, with a total of 1,000 categories. CIFAR10 and CIFAR100 are smaller datasets in image classification tasks that are often used for algorithm testing. The CIFAR-10 dataset consists of 60,000 images in 10 classes, with 6,000 images per class. The CIFAR-100 dataset has 60,000 images divided into 100 classes, each with 600 images.

CIFAR10-DVS is an event-based neuromorphic dataset converted from CIFAR10 by scanning each image with repeated closed-loop motion in front of a Dynamic Vision Sensor (DVS). There are a total of 10,000 samples in CIFAR10-DVS, with each sample lasting 300ms. The temporal and spatial resolutions are μs and $128 \times 128$, respectively. DVS128 Gesture is an event-based gesture recognition dataset, which has the temporal resolution in μs level and $128 \times 128$ spatial resolution. It records 1342 samples of 11 gestures, and each gesture has an average duration of 6 seconds.

**Data Preprocessing.** SNNs are a kind of spatio-temporal dynamic network that can naturally deal with temporal tasks. When working with static image classification datasets, it is common practice in the field to repeatedly input the same image at each timestep. As our results in Table 3 show, multiple timesteps lead to better accuracy, but also require more training time and computing hardware requirements, as well as greater inference energy consumption.

By contrast, neuromorphic datasets (i.e., event-based datasets) can fully exploit the energy-efficient advantages of SNNs with spatio-temporal dynamics. Specifically, neuromorphic datasets are produced by event-based (neuromorphic) cameras, such as DVS [95]. Compared with conventional cameras, DVS poses a new paradigm shift in visual information acquisition, which encode the time, location, and polarity of the brightness changes for each pixel into event streams with a μs level temporal resolution. Events (spike signals with address information) are generated only when the brightness of the visual scene changes. This fits naturally with the event-driven nature of SNNs. Only when there is an event input, some spiking neurons of SNNs will be triggered to participate in the computation.

Typically, event streams are preprocessed into frame sequences as input to SNNs. Details can be referred to previous work [34].

**Experimental Steup.** The experimental setup in this work generally follows [20]. The experimental settings of ImageNet-1K have been given in the main text. Here we mainly give the network settings on four small datasets. As shown in Table 5, we employ timesteps $T = 4$ on static CIFAR-10 and CIFAR-100, and $T = 16$ on neuromorphic CIFAR10-DVS and Gesture. The training epoch for these four datasets is 200. The batch size is 32 for CIFAR10/100, 16 for Gesture and CIFAR10-DVS. The learning rate is initialized to 0.0005 for CIFAR10/100, 0.0003 for Gesture, and 0.01 for CIFAR10-DVS. All of them are reduced with cosine decay. We follow [20] to apply data augmentation on Gesture and CIFAR10-DVS. In addition, the network structures used in CIFAR-10, CIFAR-100, CIFAR10-DVS, and Gesture are: spike-driven Transformer-2-512, spike-driven Transformer-2-512, spike-driven Transformer-2-256, spike-driven Transformer-2-256.

## S3    Attention Map

**Spike-Driven Self-Attention (SDSA).** Here we first briefly review the proposed spike-driven self-attention. Given a single head spike input feature sequence $S \in \mathbb{R}^{T \times N \times D}$, float-point $Q$, $K$, and $V$ in $\mathbb{R}^{T \times N \times D}$ are calculated by three learnable linear matrices, respectively. A spike neuron layer $\mathcal{SN}(\cdot)$ follows, converting float-point $Q$, $K$, $V$ into spike tensor $Q_S$, $K_S$, and $V_S$. Spike-driven self-attention is presented as:

$$\hat{V}_S = \text{SDSA}(Q, K, V) = g(Q_S, K_S) \otimes V_S = \mathcal{SN}\left(\text{SUM}_c\left(Q_S \otimes K_S\right)\right) \otimes V_S, \quad \text{(S3)}$$

where $\otimes$ is the Hadamard product, $g(\cdot)$ is used to compute the attention map, $\text{SUM}_c(\cdot)$ represents the sum of each column. The outputs of both $g(\cdot)$ and $\text{SUM}_c(\cdot)$ are $D$-dimensional row vectors. The Hadamard product between spike tensors is equivalent to the mask operation. We denote the output of $\text{SDSA}(Q, K, V)$ as $\hat{V}_S$.

Self-attention mechanism allows the model to capture long-range dependencies by attending to relevant parts of the input sequence regardless of the distance between them. In Eq. S3, SDSA adopts hard attention. The output of attention map $g(Q_S, K_S)$ is a vector containing only 0 or 1. Therefore, the whole spike-driven self-attention can be understood as masking unimportant channels in the Value tensor $V_S$. Note, instead of scale and softmax operations, we exploit Hadamard product, column element sum, and spiking neuron layer to generate binary attention scores. $Q_S$ and $K_S$ are very sparse (typically less than 0.01, see Table S1), the value of summing $Q_S \otimes K_S$ column by column does not fluctuate much, thus the scale operation is not needed here.

**Attention Map.** In a spike-driven self-attention layer, the $V_S$ and $\hat{V}_S$ of $T$ timesteps and $H$ heads are averaged. The new $V_S$ and $\hat{V}_S$ output is the spike firing rate, which we plot in Fig. S1. This allows us to observe how the attention score modulates spike firing.

Table S1: Spike Firing Rates in Spike-driven Transformer-8-512.

| | | | $T = 1$ | $T = 2$ | $T = 3$ | $T = 4$ | Average |
|---|---|---|---|---|---|---|---|
| SPS | | Conv1 | 0.0665 | 0.1260 | 0.1004 | 0.1451 | 0.1095 |
| | | Conv2 | 0.0465 | 0.0689 | 0.0597 | 0.0541 | 0.0573 |
| | | Conv3 | 0.0333 | 0.0453 | 0.0368 | 0.0394 | 0.0387 |
| | | Conv4 | 0.0948 | 0.1864 | 0.1792 | 0.1885 | 0.1622 |
| Block 1 | SDSA | Input | 0.2873 | 0.3590 | 0.3630 | 0.3625 | 0.3430 |
| | | $V_S$ | 0.2629 | 0.3094 | 0.3011 | 0.3104 | 0.2959 |
| | | $Q_S$ | 0.0142 | 0.0202 | 0.0218 | 0.0219 | 0.0195 |
| | | $K_S$ | 0.0144 | 0.0227 | 0.0234 | 0.0246 | 0.0213 |
| | | $g(Q_S, K_S)$ | 0.0792 | 0.1143 | 0.1294 | 0.1328 | 0.1139 |
| | | Output of SDSA$(\cdot)$, $\hat{V}_S$ | 0.0297 | 0.0414 | 0.0456 | 0.0508 | 0.0419 |
| | MLP | Layer 1 | 0.3675 | 0.4263 | 0.4505 | 0.4555 | 0.4250 |
| | | Layer 2 | 0.0463 | 0.0532 | 0.0520 | 0.0541 | 0.0514 |
| Block 2 | SDSA | Input | 0.3493 | 0.4002 | 0.4320 | 0.4391 | 0.4051 |
| | | $V_S$ | 0.2582 | 0.2761 | 0.2476 | 0.2237 | 0.2514 |
| | | $Q_S$ | 0.0147 | 0.0191 | 0.0195 | 0.0190 | 0.0181 |
| | | $K_S$ | 0.0128 | 0.0172 | 0.0186 | 0.0199 | 0.0171 |
| | | $g(Q_S, K_S)$ | 0.1033 | 0.1347 | 0.1357 | 0.1202 | 0.1235 |
| | | Output of SDSA$(\cdot)$, $\hat{V}_S$ | 0.03318 | 0.04373 | 0.03913 | 0.0324 | 0.0371 |
| | MLP | Layer 1 | 0.3484 | 0.3944 | 0.4259 | 0.4340 | 0.4007 |
| | | Layer 2 | 0.0317 | 0.0404 | 0.0417 | 0.0433 | 0.0393 |
| Block 3 | SDSA | Input | 0.3454 | 0.3890 | 0.4240 | 0.4292 | 0.3969 |
| | | $V_S$ | 0.3018 | 0.3055 | 0.2614 | 0.2193 | 0.2720 |
| | | $Q_S$ | 0.0108 | 0.0151 | 0.0158 | 0.0160 | 0.0144 |
| | | $K_S$ | 0.0113 | 0.0152 | 0.0151 | 0.0144 | 0.0140 |
| | | $g(Q_S, K_S)$ | 0.1273 | 0.1600 | 0.1569 | 0.1375 | 0.1454 |
| | | Output of SDSA$(\cdot)$, $\hat{V}_S$ | 0.0446 | 0.0562 | 0.0462 | 0.0344 | 0.0453 |
| | MLP | Layer 1 | 0.3436 | 0.3825 | 0.4147 | 0.4203 | 0.3903 |
| | | Layer 2 | 0.0261 | 0.0334 | 0.0347 | 0.0352 | 0.0323 |
| Block 4 | SDSA | Input | 0.3458 | 0.3855 | 0.4191 | 0.4283 | 0.3947 |
| | | $V_S$ | 0.2112 | 0.2241 | 0.1941 | 0.1728 | 0.2005 |
| | | $Q_S$ | 0.0062 | 0.0101 | 0.0113 | 0.0117 | 0.0099 |
| | | $K_S$ | 0.0061 | 0.0095 | 0.0107 | 0.0120 | 0.0096 |
| | | $g(Q_S, K_S)$ | 0.0762 | 0.0979 | 0.0981 | 0.0967 | 0.0922 |
| | | Output of SDSA$(\cdot)$, $\hat{V}_S$ | 0.0214 | 0.0289 | 0.0245 | 0.0220 | 0.0242 |
| | MLP | Layer 1 | 0.3460 | 0.3837 | 0.4146 | 0.4228 | 0.3918 |
| | | Layer 2 | 0.0208 | 0.0258 | 0.0261 | 0.0259 | 0.0247 |

| | | | $T=1$ | $T=2$ | $T=3$ | $T=4$ | Average |
|---|---|---|---|---|---|---|---|
| Block 5 | SDSA | Input | 0.3491 | 0.3908 | 0.4228 | 0.4306 | 0.3984 |
| | | $V_S$ | 0.1493 | 0.1654 | 0.1491 | 0.1395 | 0.1508 |
| | | $Q_S$ | 0.0048 | 0.0080 | 0.0090 | 0.0093 | 0.0078 |
| | | $K_S$ | 0.0042 | 0.0071 | 0.0081 | 0.0082 | 0.0069 |
| | | $g(Q_S, K_S)$ | 0.0473 | 0.0698 | 0.0740 | 0.0749 | 0.0665 |
| | | Output of SDSA($\cdot$), $\hat{V}_S$ | 0.0102 | 0.0169 | 0.0157 | 0.0147 | 0.0144 |
| | MLP | Layer 1 | 0.3541 | 0.3935 | 0.4231 | 0.4302 | 0.4002 |
| | | Layer 2 | 0.0169 | 0.0205 | 0.0205 | 0.0206 | 0.0196 |
| Block 6 | SDSA | Input | 0.3614 | 0.3957 | 0.4201 | 0.4258 | 0.4007 |
| | | $V_S$ | 0.0729 | 0.0791 | 0.0767 | 0.0778 | 0.0766 |
| | | $Q_S$ | 0.0012 | 0.0021 | 0.0027 | 0.0032 | 0.0023 |
| | | $K_S$ | 0.0008 | 0.0018 | 0.0024 | 0.0026 | 0.0019 |
| | | $g(Q_S, K_S)$ | 0.0128 | 0.0227 | 0.0260 | 0.0286 | 0.0225 |
| | | Output of SDSA($\cdot$), $\hat{V}_S$ | 0.0018 | 0.0040 | 0.0043 | 0.0045 | 0.0036 |
| | MLP | Layer 1 | 0.3690 | 0.4027 | 0.4264 | 0.4317 | 0.4074 |
| | | Layer 2 | 0.0147 | 0.0180 | 0.0183 | 0.0186 | 0.0174 |
| Block 7 | SDSA | Input | 0.3619 | 0.4069 | 0.4192 | 0.4218 | 0.4025 |
| | | $V_S$ | 0.0379 | 0.0359 | 0.0371 | 0.0406 | 0.0379 |
| | | $Q_S$ | 0.0001 | 0.0002 | 0.0003 | 0.0004 | 0.0003 |
| | | $K_S$ | 0.0001 | 0.0003 | 0.0004 | 0.0005 | 0.0003 |
| | | $g(Q_S, K_S)$ | 0.0022 | 0.0046 | 0.0058 | 0.0073 | 0.0050 |
| | | Output of SDSA($\cdot$), $\hat{V}_S$ | 0.0001 | 0.0005 | 0.0005 | 0.0006 | 0.0004 |
| | MLP | Layer 1 | 0.3575 | 0.4035 | 0.4156 | 0.4180 | 0.3987 |
| | | Layer 2 | 0.0140 | 0.0184 | 0.0186 | 0.0189 | 0.0175 |
| Block 8 | SDSA | Input | 0.2865 | 0.3888 | 0.4019 | 0.4106 | 0.3720 |
| | | $V_S$ | 0.0200 | 0.0342 | 0.0380 | 0.0419 | 0.0335 |
| | | $Q_S$ | 0.00001 | 0.0001 | 0.0001 | 0.0002 | 0.0001 |
| | | $K_S$ | $1e^{-5}$ | $1e^{-5}$ | 0.0001 | 0.0001 | $1e^{-5}$ |
| | | $g(Q_S, K_S)$ | $2e^{-5}$ | 0.0001 | 0.0020 | 0.0024 | 0.0015 |
| | | Output of SDSA($\cdot$), $\hat{V}_S$ | $1e^{-5}$ | 0.0002 | 0.0002 | 0.0002 | 0.0001 |
| | MLP | Layer 1 | 0.2716 | 0.3721 | 0.3827 | 0.3899 | 0.3541 |
| | | Layer 2 | 0.0056 | 0.0111 | 0.0110 | 0.0115 | 0.0098 |
| Head | FC | | 0.0002 | 0.3876 | 0.3604 | 0.4843 | 0.3081 |

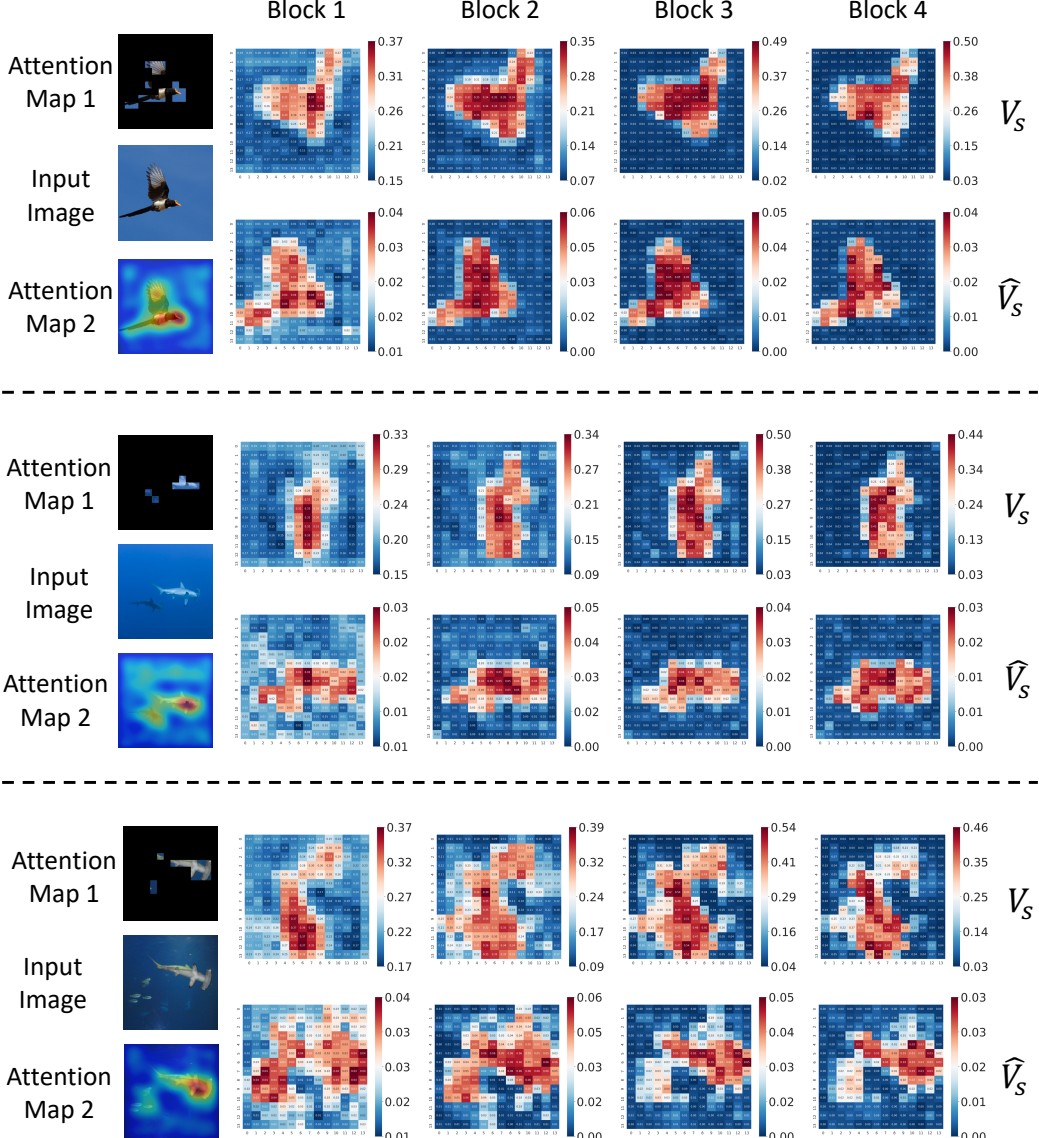

Figure S1: Attention Map Based on Spike Firing Rate (SFR). Attention map 1 and 2 are generated by the Grad-CAM method [96]. $V_S$ is the Value tensor. $\hat{V}_S$ is the output of $\mathrm{SDSA}(\cdot)$. The spike-driven self-attention mechanism masks unimportant channels in $V_S$ to obtain $\hat{V}_S$. Each pixel on $V_S$ and $\hat{V}_S$ represents the SFR at a patch. The spatial resolution of each attention map is $14 \times 14$ (196 patches). The redder the higher the SFR, the bluer the smaller the SFR. We can see that the $\mathrm{SDSA}(\cdot)$ regulation of spike firing is basically consistent with the focused points in the attention map.