# OpenReview forum: "Spike-driven Transformer"
_NeurIPS.cc/2023/Conference — NeurIPS 2023 poster_

### Official Review · Reviewer_qknk · 2023-07-06

**Soundness:** 2 fair
**Presentation:** 3 good
**Contribution:** 3 good
**Rating:** 5
**Confidence:** 4

**Summary:**

This paper proposes a Spike-driven Transformer that incorporates the spike-driven paradigm of SNNs into the Transformer and is hardware-friendly for neuromorphic chips. The authors use SDSA to transform the multiplication operation between Query, Key, and Value into a combination of mask and sparse addition operations and modify the self-attention structure to achieve linear complexity. As a result, this approach can achieve up to 87.2× lower energy consumption compared to the traditional self-attention method. Furthermore, membrane shortcut is introduced to ensure all spiking neurons communicate via binary spikes. The paper provides insights into implementing each module on neuromorphic chips and achieves competitive results on multiple datasets while reducing energy consumption compared to previous works.

**Strengths:**

The authors introduce the Spike-Driven Self-Attention (SDSA) module as a replacement for the self-attention module in the current spiking Transformer. The proposed module exhibits superior performance while significantly reducing energy consumption, resulting in an architecture that is better suited for deployment on neuromorphic chips.

**Weaknesses:**

This paper explores the feasibility of implementing Conv, MLP, and Self-attention models driven by spikes on neuromorphic chips. However, it does not delve into the implementation details of membrane shortcuts. This approach involves the direct transmission of membrane potential signals between spiking neurons and may contradict the spike-driven approach discussed in the original text.

The effectiveness of a spikformer-based framework with only SDSA, without MS, remains unclear as most experiments in the literature combine both techniques. This weakens the argument for the performance of the spike-driven Transformer proposed in this paper. Although an ablation study was conducted on MS and SDSA in Table 5, it was limited to the CIFAR10/100 dataset and Spiking Transformer-2-512. The results suggest that the use of MS may be crucial to achieve state-of-the-art performance.


**Questions:**

How is the membrane shortcut implemented on a neuromorphic chip?  Is it necessary to use it instead of SEW shortcut? The article explains that membrane shortcuts are used to optimize the distribution of membrane potential, but for spiking neurons, wouldn't it be more appropriate to perform such an optimization in the time domain?

---

> ### Author Rebuttal · Authors · 2023-08-06
>
> Thank you for your insightful feedback. We have carefully studied your comments and argue that **your concerns can be addressed**.
>
> We incorporate the spike-driven paradigm into Transformer. In spike-driven Transformer, there are only sparse additions. This limitation is quite severe, but we still achieved SOTA on ImageNet in SNN. To achieve this, we designed a novel Spike-Driven Self-Attention (SDSA) and tuned shortcuts. **What's more, the SDSA operator is the third class of operators besides spike-driven Conv and MLP, which will inspire future neuromorphic chip designs.** Since this work is the first to incorporate spike-driven into Transformer, it is reasonable to think that it has more room for improvement.
>
> **This work will undoubtedly impact the SNN domain in algorithm and neuromorphic chip design. Given the significant technical contributions and importance of this work to the field, we sincerely hope that you will reconsider your rating**
>
> > *W1 and Q1*. it does not delve into the implementation details of Membrane Shortcuts (MS). This approach... How is MS implemented on a neuromorphic chip?
>
> **A: The MS does not conflict with spike-driven**
>
> - **Spike-driven** implies spike-based event-driven, which restricts all multiplications associated with spike tensors to be implemented as additions.
>
> - **Spike-driven on a neuromorphic chip.** Spike-driven paradigm is implemented on neuromorphic chips in the form of **addressable algorithms[1]**. The standard protocol for spike communication is the asynchronous Address-Event Representation (AER), from simple point-to-point links to complex networks-on-a-chip. So, spike-driven router occupies an important position in chip design.
>
> - **MS on a neuromorphic chip.** The design of a spike-driven router is affected by many factors, e.g., chip architecture, layout, clock, manufacturing process, applications. Let's take Speck of SynSense as an example to illustrate how to implement MS. Speck is an asynchronous chip focused on processing event streams. In Speck [2], Whenever an event arrives at an SNN core with its address information, the corresponding kernel value and destination neuron position are obtained by address searching, the destination neuron states are then asynchronously updated according to the synaptic operation. And, asynchronous spike-driven convolution is independent to the arrival of other input events and cores, the operation can be efficiently parallel distributed for multiple events at different positions. Specific to using MS in Speck, when a spiking neuron receives a spike, the membrane potential must change. Then, another addressing function can be used to pass this membrane potential to the corresponding neuron in the subsequent layer for merging.
>
> > *Q2*. Is it necessary to use MS shortcut instead of SEW shortcut?
>
> **A:** It depends on the specific situation. We give a very strict restriction that there can only be the addition in the whole Transformer. In this case, the SEW cannot exist because it would bring integer multiplication. If we relax this restriction or there are neuromorphic chips that support integer multiplication, SEW can also exist.
>
> > *Q3*. The article explains that membrane shortcuts are used to optimize the distribution of membrane potential, but for spiking neurons, wouldn't it be more appropriate to perform such an optimization in the time domain?
>
> **A:** It is indeed feasible to perform membrane potential optimization in the time domain on some small time-domain tasks, e.g., using long short-term memory[3]. But when processing large static datasets, such as ImageNet, where deep SNNs are required, membrane potential optimization using residual connections in the spatial-domain to avoid performance degradation is mandatory[4].
>
> > *W2*. The effectiveness of a spikformer-based framework with only SDSA, without MS, remains unclear... This weakens the argument for the performance of the spike-driven Transformer. The results suggest that the use of MS may be crucial to achieve SOTA.
>
> **A:** We tend to understand spike-driven Transformer from the view of MetaFromer[6], which argues that there is general architecture abstracted from Transformers by not specifying the token mixer. Setting the operators in the token mixer will result in different accuracies. So, MS and SEW are changes to the architecture, while SDSA and SSA are adjustments to operators. To verify this, we set $T=4$, structure-8-384 on ImageNet as follows:
>
> |Model|SEW+SSA(Base[5])|SEW+SDSA|MS+SSA|MS+SDSA(This work)|
> |---|---|---|---|---|
> |Acc(\%)|70.2|68.1|72.7|72.3|
>
> We can see that MS brings gain to architecture performance, MS+SSA(72.7) vs. SEW+SSA(70.2), and MS+SDSA(72.3) VS. SEW+SDSA(68.1). And, compared to SSA, SDSA indeed does perform worse. But it should be noted that there is only addition in the proposed SDSA and the complexity is $O(ND)$, while the SSA in [5] contains multiplication (multi-bit integer multiplication and scale operations) and the complexity is $O(ND^2)$. Note, SDSA is a severely restricted operator, but it works pretty well. So, once the basic architecture is determined, the operator has a trade-off between accuracy and energy.
>
> We appreciate that you can discuss this issue with us. **This actually points out two directions for further optimization: architecture and spike-driven operators.**
>
> ---
> [1] Bottom-Up and Top-Down Approaches for the Design of Neuromorphic Processing Systems: Tradeoffs and Synergies Between Natural and Artificial Intelligence. In Proceedings of the IEEE, 2023.
>
> [2] Event-driven spiking convolutional neural network. WIPO Patent, page WO2020207982A1, 2020.
>
> [3] A long short-term memory for AI applications in spike-based neuromorphic hardware. In Nature Machine Intelligence, 2022.
>
> [4] Attention Spiking Neural Networks. In IEEE T-PAMI, 2023.
>
> [5] Spikformer: When Spiking Neural Network Meets Transformer. In ICLR, 2023.
>
> [6] Metaformer is actually what you need for vision. In CVPR 2022.

---

> > ### Comment · Reviewer_qknk · 2023-08-13
> >
> > Thank you for providing detailed responses and conducting additional experiments in your rebuttal. It is evident that SDSA can significantly reduce computational complexity without using MS, while maintaining competitive performance. Moreover, this approach facilitates easy deployment on neuromorphic chips. However, I still hold reservations regarding the use of voltage as output for membrane shortcut, as it does not align with the mechanisms of SNNs.
> > In summary, I am willing to revise my score from 4 to 5.

---

> > > ### Author Response · Authors · 2023-08-17
> > >
> > > Thank you very much for your endorsement, the discussion with you made us rethink this work carefully. We believe this is very helpful for our future work.

---

### Official Review · Reviewer_8FkB · 2023-07-06

**Soundness:** 3 good
**Presentation:** 3 good
**Contribution:** 3 good
**Rating:** 7
**Confidence:** 3

**Summary:**

This paper proposed a Spike-Driven Self-Attention (SDSA) module, which uses Hadamard product, column-wise summation, and spiking neuron layer to replace the matrix multiplication and softmax operation. Experiments on static and neuromorphic image classification demonstrate competitive performance and energy efficiency.

**Strengths:**

1.. The authors proposed a novel form of linear self-attention module, SDSA, which leverages the features of the spikes and increases computational and memory efficiency.
2. The authors show extensive results from different models and different datasets. The performance of accuracy and energy efficiency is very strong.
3. SDSA reduces the computational complexity with a slight accuracy drop, and the authors show some attention maps to validate the effectiveness of the SDSA.

**Weaknesses:**

1. There is still a noticeable drop between the spike-driven transformer and the original transformer. Can the authors show some results of the model with more time steps and report the limitations of the model?
2. Can the authors show the performance and energy consumption as the number of time steps decreases?

**Questions:**

1. The ablation study shows the membrane shortcut brings significant accuracy improvement. I slightly doubt the importance of the SDSA. Can the authors try the combination of the membrane shortcut and some attention-free transformers like [1][2] to check if they can also achieve similar performance?


[1] J Lee-Thorp et al., “FNet: Mixing Tokens with Fourier Transforms”, NAACL 2022
[2] W Yu et al, “MetaFormer Is Actually What You Need for Vision”, CVPR 2022

**Limitations:**

The paper has few limitations.

---

> ### Author Rebuttal · Authors · 2023-08-06
>
> Thank you for your insightful comments. We first list your advice and questions, then give our detailed answers.
>
> > *Weakness 1 and 2:* There is still a noticeable drop between the spike-driven transformer and the original transformer. Can the authors show some results of the model with more time steps and report the limitations of the model? Can the authors show the performance and energy consumption as the number of time steps decreases?
>
> **A:** Generally, in directly training SNNs, for image classification, the larger the timestep, the higher the performance and inference energy cost, and the training cost also increases rapidly. However, as the timestep increases, the performance gain becomes smaller and smaller. Therefore, this is a trade-off problem. In existing deep SNNs [3,4,5], researchers generally exploit $T=4$ by default. Due to time and resource constraints, We only tested up to $T=6$ on spiking Transformer-384.
>
> | Timestep | $T=1$ | $T=2$ | $T=4$ | $T=6$|
> |---|---|---|---|---|
> | Acc (\%) | 71.7 | 72.9 | 74.6 | 74.8 |
> | Power (mJ) | 1.13 | 2.33 | 4.50 | 7.17 |
>
> **A major limitation of SNNs is that, although SNNs are theoretically highly energy-efficient, they often require specialized hardware (such as neuromorphic chips [6]) for verification.** Thus, SNNs are in fact a sparse computing paradigm that requires algorithm-hardware co-design. This is why we believe that this paper will have a profound impact on the SNN field. **First, our model raises the performance ceiling in the field of SNNs**. Given the fact that this work is the first to incorporate spike-driven into Transformer, it is reasonable to think that it has more room for improvement. **On the other hand, to the best of our knowledge, the proposed Spike-Driven Self-Attention (SDSA) operator is the third class of existing operators besides spike-driven Conv and MLP. So, we think our work will also have a profound impact on future neuromorphic chip design.**
>
> > *Question:* The ablation study shows the membrane shortcut brings significant accuracy improvement. I slightly doubt the importance of the SDSA. Can the authors try the combination of the membrane shortcut and some attention-free transformers like [1][2] to check if they can also achieve similar performance?
>
> **A:** **We would be happy to discuss this with you. MetaFormer has inspired us a lot, and we understand your idea.** We organized some experiments, set $T=1$, 100epoch, and spiking Transformer 8-384.The experimental results are as follows:
>
> | Model | Baseline(SDSA, This work) | +SSA [7] | +Fourier [1] | +Pooling [2] |
> |---|---|---|---|---|
> | Acc (\%) | 61.0 | 63.7 | Not convergent | 41.2 |
>
> We suspect that the performance gap between Baseline(SDSA) and +SSA is due to insufficient training. Therefore, we set $T=4$ and 310epoch and retrain. The experimental results are as follows:
>
> | Model | Baseline(SDSA, This work) | +SSA [6] |
> |---|---|---|
> | Acc (\%) | 72.3 | 72.7 |
>
> According to our understanding of MetaFormer and the above experimental results, the following preliminary conclusions can be drawn:
>
> (1) **We think that spike-driven Transformer can be considered as a kind of MetaFormer (CAFormer in [8], Conv + ViT). The proposed SDSA is a token mixer.** In this perspective, when we replace the token mixer from SDSA (this work) to SSA[7], the accuracy of both $T=1$ and $T=4$ will improve. However, it should be noted that there is only binary spike-based addition in the proposed SDSA and the computational complexity is $O(ND)$, while the SSA in [7] contains multiplication (multi-bit integer multiplication and scale operations) and the computational complexity is $O(ND^2)$.
>
> (2) **Directly employing the Fourier or Pooling operator to replace SDSA as the token mixer does not work well.** The reason has not yet been confirmed. But we think it will be a good perspective to understand and improve spike-driven Transformer in the future.
>
> ---
> [1] FNet: Mixing Tokens with Fourier Transforms, In NAACL, 2022.
>
> [2] MetaFormer Is Actually What You Need for Vision. In CVPR 2022.
>
> [3] Deep Residual Learning in Spiking Neural Networks. In NeurIPS, 2021.
>
> [4] Temporal efficient training of spiking neural network via gradient re-weighting. In ICLR 2022.
>
> [5] Attention Spiking Neural Networks. In IEEE T-PAMI, 2023.
>
> [6] Towards artificial general intelligence with hybrid Tianjic chip architecture, In Nature, 2019.
>
> [7] Spikformer: When Spiking Neural Network Meets Transformer. In ICLR, 2023.
>
> [8] Metaformer baselines for vision. In arXiv 2022.

---

> > ### Comment · Reviewer_8FkB · 2023-08-21
> > **Thank you for addressing our questions**
> >
> > Spike Transformer: The authors have addressed our questions with more experimental results. We have raised our score to 7.

---

> > > ### Author Response · Authors · 2023-08-21
> > >
> > > Thank you so much for your endorsement, your comments have inspired us and we are happy to discuss it with you.

---

### Official Review · Reviewer_Lu4B · 2023-07-07

**Soundness:** 3 good
**Presentation:** 2 fair
**Contribution:** 4 excellent
**Rating:** 7
**Confidence:** 4

**Summary:**

This paper proposes an improved spike transformer by replacing Spike-Element-Wise shortcut in an existing spike transformer (ref [20]) with Membrane Shortcut from spike ResNet (ref [26]) .

**Strengths:**

Match SOTA ImageNet top-1 precision achieved by ResNet [26] with slightly lower estimated energy consumption.

**Weaknesses:**

Combining known techniques from two papers without modification is considered an incremental improvement.
It is not clear why Transformer is a better choice than ConvNet in the context of Spike Neural Network. In ViT papers, the motivation for the Transformer is its model capacity can scale easily to handle large datasets and it has much better parallelism on TPU/GPU than ConvNet. None of these has been indicated as the goal for SNN. As shown in table-2, the ResNet SNN model in ref[27] is equally competitive as the proposed model.
The notation is a bit confusing in a few equations. The s in Eq (5) is binary. This should be distinguished from floating-point numbers in Eq (4). But the same symbol R is used for both. The same observation holds for Eq (8)(10)(11).

**Questions:**

The outputs of LIF spiking neurons at all timesteps are used as the inputs to the Transformer. This may be the tested method in [20], but why is this a better idea than using the output at the last timestep? As shown in table-2, the power is directly proportional to the timestep count. So this seems a critical parameter to optimize over.

In Eq (12), S_L has the shape TxNxD. So outputs at all timesteps are used in classification head?

In Eq (6), the input s has no 2d spatial dimensions. What are the 2d dimensions in Conv2d()?

---

> ### Author Rebuttal · Authors · 2023-08-06
>
> Thank you for your insightful feedback. We first list your advice and questions, then give our detailed answers. **And, we really hope that you would re-consider your rating, given this work's notable contribution to the SNN field.**
>
> > *Summary and Weakness 1*: This paper proposes an improved spike transformer by replacing Spike-Element-Wise shortcut in an existing spike transformer (ref [20]) with Membrane Shortcut from spike ResNet (ref [26]). Combining known techniques from two papers without modification is considered an incremental improvement.
>
> **A**: We would like to discuss the contribution and significance of this work with you in depth.
>
> (1) **Goals and status quo in SNN.** The ambition of SNNs is to be a low-power alternative to ANNs. The key to low-power of SNN is the **spike-driven**. There are **two main hurdles** in achieving this goal: First, there is a performance gap between SNNs and ANNs; Second, the low-power of SNNs can only be truly realized when run on a neuromorphic chip, thus SNNs are in fact a computing paradigm that requires algorithm-hardware co-design.
>
> (2) **Technical Contributions.** We incorporate the spike-driven paradigm into Transformer by the proposed Spike-driven Transformer.
>
> - **SDSA operator.** There are only two types of operators in SNN, spike-driven Conv and MLP. **SDSA is the first spike-driven Self-attention operator** that implements the self-attention using mask and addition without any multiplication (e.g., softmax and scale). SDSA is computationally linear in both tokens and channels. Its energy cost is $87.2\times$ lower than vanilla self-attention.
>
> - **Rearrange shortcuts.** We use membrane potential shortcuts to avoid integer (multi-bit spikes).
>
> - **Spike-driven Transformer** has only sparse addition and achieves SOTA performance on ImageNet.
>
> (3) **Significance.**
>
> - **Energy efficiency.** Compared with the prior SNNs with the same parameter amount, spike-driven Transformer has higher performance but lower energy consumption. Compared with the ANN counterpart, the energy efficiency of spike-driven Transformer is as high as $36.7 \times$.
>
> - **Performance.** Achieved a breakthrough from 0 to 1, we incorporated the spike-driven paradigm into Transformer and achieved SOTA results. There is huge room for future performance improvement.
>
> - **Algorithms drive the hardware design.** SDSA is the first spike-driven self-attention operator, which is expected to advance the design of next-generation neuromorphic chips.
>
> **Thus, the contribution of this work is by no means a mere substitution of shortcuts on the basis of prior work. We can confidently say that this work will change the field of SNNs in terms of algorithm and neuromorphic chip design.**
>
> > *Weakness 2*: It is not clear why Transformer is a better choice than ConvNet in the context of Spike Neural Network. In ViT papers, the motivation for the Transformer is its model capacity can scale easily to handle large datasets and it has much better parallelism on TPU/GPU than ConvNet. None of these has been indicated as the goal for SNN. As shown in table-2, the ResNet SNN model in ref[27] is equally competitive as the proposed model.
>
> **A**: This is an important issue. In fact, the proposed architecture is a Conv and ViT hybrid network to extract information in a local-global manner. This is somewhat similar to MetaFormer.
>
> - **MetaFormer[1,2]** argues that there is general architecture abstracted from Transformers by not specifying the token mixer. Setting the operators in the token mixer to Identity mapping, Conv, MLP, or Self-attention will result in different accuracies. The results show that the Self-attention, which can extract global information, can achieve the highest performance.
>
> - **Spike-driven Transformer** can be considered as a kind of MetaFormer (CAFormer in [2], Conv + ViT). The proposed novel SDSA is a token mixer.
>
> We argue that from the perspective of MetaFormer, SDSA operators bring global information to the network based on the infrastructure. Interestingly, the attention SNN in ref[27] also adds a global attention module on vanilla Res-SNN. In contrast to Multiply-and-Accumulate attention module in ref[27], the global SDSA operator is purely additive. Given the fact that this work is the first to incorporate spike-driven into Transformer, it is reasonable to think that it has more room for improvement.
>
> > *Weakness2,Q3*: The notation is a bit confusing in a few equations. The s in Eq (5) is binary. This should be distinguished from floating-point numbers in Eq (4). But the same symbol R is used for both. The same observation holds for Eq (8)(10)(11). In Eq (6), the input s has no 2d spatial dimensions. What are the 2d dimensions in Conv2d()?
>
> **A**: We apologize for the confusion caused by the imprecise notation. In Eq.(6), before executing $Conv2d(\cdot)$, $s\in \mathbb{R}^{T\times N\times D}$ will be transposed into $s\in \mathbb{R}^{T\times C\times H \times W}$.
>
> > *Q1,2*: The outputs of LIF spiking neurons at all timesteps are used as the inputs to the Transformer. This may be the tested method in [20], but why is this a better idea than using the output at the last timestep? As shown in table-2, the power is directly proportional to the timestep count. So this seems a critical parameter to optimize over. In Eq (12), S_L has the shape TxNxD. So outputs at all timesteps are used in classification head?
>
> **A**: Yes, in SNN, it is a default operation to use the output of all timesteps for classification [3], because it is more accurate. Generally, for image classification, the larger the timestep, the higher the performance and inference energy cost, and the training cost also increases rapidly. Therefore, this is a trade-off problem.
>
> ---
> [1] Metaformer is actually what you need for vision. In CVPR 2022.
>
> [2] Metaformer baselines for vision. In arXiv 2022.
>
> [3] Temporal efficient training of spiking neural network via gradient re-weighting. In ICLR 2022.

---

> ### Comment · Reviewer_Lu4B · 2023-08-20
>
> Thanks for the clarification. I have increased my ratings.

---

> > ### Author Response · Authors · 2023-08-21
> >
> > I sincerely appreciate your constructive comments. But I noticed that you don't seem to be improving the rating of this paper as you said in official comment. I wonder if there was some misunderstanding here, could you please check it again?

---

### Official Review · Reviewer_GiWX · 2023-07-07

**Soundness:** 3 good
**Presentation:** 3 good
**Contribution:** 3 good
**Rating:** 7
**Confidence:** 2

**Summary:**

The authors propose a variant of transformer networks with spiking neurons based on the LIF neuron. The submission reformulates the self-attention to use sparse addition and masking, and modifies the residual connections to transmit information in the domain of membrane potentials.
They achieve a state of the art result on Imagenet.
They present an energy analysis.


**Strengths:**

It increases the amount of binary spike-based computations in the transformer.
The experimental results are strong.
A small ablation study is performed.

**Weaknesses:**

The explanation for  equations 15 and 16 can be improved
The explanation of power estimates can be improved.

**Questions:**

Can the authors provide an additional explanation for equations 15 and 16?

**Limitations:**

not much discussed

---

> ### Author Rebuttal · Authors · 2023-08-05
>
> Thanks for your insightful feedback and your time in reading our paper. Due to space constraints in the paper, the explanations for Eq.(15) and (16) are rough, and we put some details of energy consumption evaluation in the Supplementary Material. After careful inspection, we found that there is a typo in Eq.(16), we accidentally wrote $V^{i}$ instead of $Q^{i}$. We're sorry for this and hope that the responses below can answer your concerns.
>
> > *Question*: Can the authors provide an additional explanation for equations 15 and 16?
>
> **A**: Here we first briefly introduce three typical attention mechanisms in existing Transformers from the perspective of computational complexity: Vanilla Self-Attention(VSA) [1], Linear Attention [2], Hydra Attention [3]. Then, we introduce the proposed Spike-Driven Self-Attention (SDSA) and analyze its computational complexity.
>
> ***
> We can understand self-attention from the perspective of computational complexity. Specifically, two matrix multiplications between float-point $Q$, $K$, $V$ in $\mathbb{R}^{N\times D}$ are included in the VSA, where $N$ is the token number, $D$ is the channel dimension. Generally, VSA performs multi-head self-attention, i.e., divide $Q$, $K$, $V$ into $H$ heads in the channel dimension. In the $i$-th head, $Q^{i}$, $K^{i}$, $V^{i}$ in $\mathbb{R}^{N \times D/H}$. After the self-attention operation is performed on the $H$ heads respectively, the outputs are concatenated together.
>
> (1) Vanilla self-attention [1]. $Q$ and $K$ are matrix multiplied first, and then their output is matrix multiplied with $V$. The computational complexity of VSA is $O(N^2D)$, which has a **quadratic** relationship with the toke number $N$.
>
> (2) Linear attention [2,3]. $K$ and $V$ are matrix multiplied first, and then their output is matrix multiplied with $Q$. The computational complexity of linear attention is $O(ND^2/H)$, which has a **linear** relationship with the toke number $N$.
>
> (3) Hydra attention [3]. Consider an extreme case in linear attention, set $H=D$. That is, in each head, $Q^{i}$, $K^{i}$, $V^{i}$ in $\mathbb{R}^{N \times 1}$. Then the computational complexity of hydra attention is $O(ND)$, which has a **linear** relationship with both the toke number $N$ and the channel dimension $D$.
>
> ***
> (4) **Spike-driven self-attention (This work)**. We show that SDSA has the same computational complexity as hydra attention (Lines179-184), i.e., $O(ND)$. We here assume $T=1$ for mathematical understanding. Note, in SNN, a spike neuron layer $SN(\cdot)$ first converts $Q$, $K$, $V$ into spike tensor $Q_{S}$, $K_{S}$, and $V_{S}$. The proposed SDSA Version 1 (SDSA-V1) is executed as:
>
> $SDSA(Q, K, V)$ = $g(Q_{S}, K_{S}) \otimes V_{S}$ = $SN(SUM_{c}(Q_{S} \otimes  K_{S})) \otimes V_{S}$ (14)
>
> where $\otimes$ is the Hadamard product, $g(\cdot)$ is used to compute the attention map, $SUM_{c}(\cdot)$ represents the sum of each column. The outputs of both $g(\cdot)$ and $SUM_{c}(\cdot)$ are $D$-dimensional row vectors. The Hadamard product between spike tensors is equivalent to the mask operation. **Since the Hadamard product among $Q_{S}$, $K_{S}$, and $V_{S}$ can be exchanged**, Eq.(14) can also be written as (SDSA-V2):
>
> $SDSA(Q, K, V)$ = $Q_{S} \otimes g(K_{S}, V_{S})$ = $Q_{S} \otimes SN(SUM_{c}(K_{S} \otimes  V_{S}))$ (15)
>
> In Eq.(15), $K_{S}$ and $V_{S}$ participate in the operation first, **thus it is a kind of linear attention. Further, we consider the special operation of Hadamard product.** Specifically, the output of $SUM_{c}(K_{S} \otimes  V_{S})$ is a $D$-dimensional row vector, and the value of the $i$-th element in $SUM_{c}(K_{S} \otimes  V_{S})$ is $D_{i}$, then
>
> $D_{i}$ = $SUM_{c}(K_{S}^{i} \otimes  V_{S}^{i})$ = $(K_{S}^{i})^{\rm{T}}  \odot  V_{S}^{i}$ = $SN(K^{i})^{\rm{T}} \odot SN(V^{i})$,
>
> where $\odot$ is the dot product operation, $K_{S}^{i} = SN(K^{i})$ and $V_{S}^{i} = SN(V^{i})$ are the $i$-th column vectors in $K_{S}$ and $V_{S}$, respectively. **In simple terms, taking the Hadamard product of two column vectors $a$ and $b$ and summing them is equivalent to multiplying $b$ times the transpose of $a$, i.e., $SUM_{c}(a \otimes b) = a^{\rm{T}} \odot b$**. Therefore, we can get the V2 version of SDSA:
>
> $SDSA(Q^{i}, K^{i}, V^{i})$ = $SN(Q^{i})g(K^{i}, V^{i})$ = $SN(Q^{i})SN(SUM_{c}(K_{S}^{i} \otimes  V_{S}^{i}))$ = $SN(Q^{i}) SN(SN(K^{i})^{\rm{T}} \odot {SN}(V^{i}))$ (16)
>
> where $Q^{i}, K^{i}, V^{i}$ in $\mathbb{R}^{N\times 1}$ are the $i$-th vectors in $Q, K, V$ respectively. The output of $g(K^{i}, V^{i})$ is a scalar, 0 or 1. Since the operation between $SN(Q^{i})$ and $g(K^{i}, V^{i})$ is a mask, the whole SDSA only needs to calculate $D$ times for $g(K^{i}, V^{i}) = SN(D_{i})$. Note that only $N$ additions need to be performed in $D_{i}$ = $SUM_{c}(K_{S}^{i} \otimes  V_{S}^{i})$. Thus, the computational complexity of SDSA is $O(0+ND)$, which is linear with both $N$ and $D$. Vectors $K_{S}^{i}$ and $K_{S}^{i}$ are very sparse, typically less than 0.01 (see Supplementary Material). Together, the whole SDSA only needs about $0.02ND$ times of addition.
>
> ***
> [1] Attention is all you need. In: NeurIPS (2017)
>
> [2] Transformers are RNNs: fast autoregressive transformers with linear attention. In: ICML (2020)
>
> [3] Hydra attention: Efficient attention with many heads. In: ECCV (2022)

---

> > ### Comment · Reviewer_GiWX · 2023-08-16
> > **why is the right hand side in eq (16) equal to the right hand side in eq (14) ?**
> >
> > Thank you for writing the explanation, however this still raises doubts:
> >
> >
> > eq 16, rhs is:
> > $SN(Q^i) \cdot  SN  ( SN(K^i)^\top  \odot   SN(V^i)  )   $
> >
> > eq 14 rhs is:
> > $ SN  ( SN(K^i)^\top  \odot   SN(Q^i)  ) \cdot  SN(V^i)  $
> >
> >
> >
> > the hadamard product can be exchanged:
> > $Q_s \otimes K_s  \otimes V_s =  (Q_s \otimes K_s ) \otimes V_s = Q_s \otimes (K_s  \otimes V_s) $,
> >  but not when there are thresholding non-linearities applied in between.
> >
> > As far as the reviewer understands, SN is a thresholding operation resulting in 1 or 0.
> >
> > So something needs to be proven to show that
> >
> > $SN (  SUM_c Q_s \otimes K_s ) \otimes SN( V_s)  =  SN(Q_s) \otimes SN ( SUM_c K_s  \otimes V_s) $
> >
> > This could be due to the fact that the summings are all non-negative, but even then the threshold should be at 1 or below. if the threshold is above 1, then this equality might not hold anymore.
> >
> > Can you please clarify ?

---

> > > ### Author Response · Authors · 2023-08-16
> > > **Eq.(14) and Eq.(15) are functionally equivalent. Eq.(15) and Eq.(16) are mathematically equivalent**
> > >
> > > Thank you very much for your insightful comments. The expression associated with Eq.(15) is not rigorous. **Eq.(14) and Eq.(15) are indeed not equivalent mathematically, but they are equivalent functionally.**
> > >
> > > **SDSA-V1.** Given a spike input feature sequence $S \in \mathbb{R}^{T\times N\times D}$, float-point $Q$, $K$, and $V$ in $\mathbb{R}^{T\times N\times D}$ are calculated by three learnable linear matrices, respectively. A spike neuron layer $SN(\cdot)$ follows, converting $Q$, $K$, $V$ into spike tensor $Q_{S}$, $K_{S}$, and $V_{S}$. SDSA-V1 is presented as:
> > >
> > > $SDSA(Q, K, V)=g(Q_{S}, K_{S})\otimes V_{S}= SN(SUM_{c}(Q_{S} \otimes  K_{S})) \otimes V_{S},$ (14)
> > >
> > > **Since $Q_{S}, K_{S}, V_{S}$ are generated by the same type of function (a linear transformation) from the same input $X$ and there is no softmax in Eq.(14), $Q_{S}, K_{S}, V_{S}$ are no longer Query, Key, Value with clear meaning. Therefore, we can let $K_{S}$ be the Query matrix and $V_{S}$ be the Key matrix.** This is Eq.(15) below:
> > >
> > > $SDSA(Q, K, V)=Q_{S} \otimes g(K_{S}, V_{S}) = Q_{S} \otimes SN(SUM_{c}(K_{S} \otimes  V_{S}))$, (15)
> > >
> > > **In summary, Eq.(14) and Eq.(15) are functionally equivalent. Eq.(15) and Eq.(16) are mathematically equivalent (please see previous reply).** For rigor, we will revise Lines 177-178 in the main text.
> > >
> > > Original Lines 177-178: ``Since the Hadamard product among $Q_{S}$, $K_{S}$, and $V_{S}$ in $\mathbb{R}^{N\times D}$ can be exchanged, Eq.(14) can also be written as: $SDSA(Q, K, V)=Q_{S} \otimes g(K_{S}, V_{S}) = Q_{S} \otimes SN(SUM_{c}(K_{S} \otimes  V_{S}))$ "
> > >
> > > Revised Lines 177-178: ``***Since we exploit the Hadamard product and the sum function to calculate the similarity, and $Q_{S}$, $K_{S}$, and $V_{S}$ are generated in exactly the same way, Eq.(14) is functionally equivalent to the following formula***: $SDSA(Q, K, V)=Q_{S} \otimes g(K_{S}, V_{S}) = Q_{S} \otimes SN(SUM_{c}(K_{S} \otimes  V_{S}))$ "
> > >
> > > We would also like to explain to you why we have given Eq.(15) and Eq.(16). A hallmark of linear Transformer[1,2] is that Key matrix and Value matrix are computed first, followed by Query matrix. We want to give readers such a formal intuition that the proposed SDSA is a kind of linear attention. In the code implementation, there is no difference in performance between Eq.(14) and Eq.(15). Thank you again for your careful review and we will make changes accordingly.
> > >
> > > ---
> > > [1] Transformers are RNNs: fast autoregressive transformers with linear attention. In: ICML (2020)
> > >
> > > [2] Hydra attention: Efficient attention with many heads. In: ECCV (2022)

---

> > > > ### Comment · Reviewer_GiWX · 2023-08-16
> > > >
> > > > thank you for taking the time to explain it. The proposed clarification for the paper that it is functionally equivalent but not mathematically equivalent is a good idea for the paper. The reviewer is happy about it (not only because the reviewer wants to sink into sleep now).

---

> > > > > ### Author Response · Authors · 2023-08-17
> > > > >
> > > > > Thank you very much for your recognition, your comments were very inspiring and helped us to improve this paper.

---

### Decision · Program_Chairs · 2023-09-21

**Decision:**

Accept (poster)

**Comment:**

This work proposes improving the spike transformer by using the membrane shortcut from spike ResNet and a new spike-driven self-attention operator that only uses binary spike-based addition. This is used to achieve SOTA ImageNet top-1 in the SNN field while using lower estimated energy. This will likely be an impactful paper in the SNN field.

Advantages:
- SOTA results for ImageNet in the SNN field.
- Lower estimated energy consumption from the newly proposed SDSA operator.
- The new SDSA operator has lower computational complexity than SSA.

Weaknesses:
- The notation in the equations is a bit confusing.
- MS (membrane shortcut) does not align with SNN mechanisms and its potential implementation is somewhat unclear.